# Wearable Online Freezing of Gait Detection and Cueing System

**DOI:** 10.3390/bioengineering11101048

**Published:** 2024-10-20

**Authors:** Jan Slemenšek, Jelka Geršak, Božidar Bratina, Vesna Marija van Midden, Zvezdan Pirtošek, Riko Šafarič

**Affiliations:** 1Faculty of Mechanical Engineering, University of Maribor, 2000 Maribor, Slovenia; jelka.gersak@um.si; 2Faculty of Electrical Engineering and Computer Science, University of Maribor, 2000 Maribor, Slovenia; bozidar.bratina@um.si (B.B.); riko.safaric@um.si (R.Š.); 3Department of Neurology, University Clinical Center Ljubljana, 1000 Ljubljana, Slovenia; vesna.vanmidden@kclj.si (V.M.v.M.); zvezdan.pirtosek@kclj.si (Z.P.)

**Keywords:** Parkinson’s disease, freezing of gait, machine learning, real-time systems, wearable devices, on-demand stimulation

## Abstract

This paper presents a real-time wearable system designed to assist Parkinson’s disease patients experiencing freezing of gait episodes. The system utilizes advanced machine learning models, including convolutional and recurrent neural networks, enhanced with past sample data preprocessing to achieve high accuracy, efficiency, and robustness. By continuously monitoring gait patterns, the system provides timely interventions, improving mobility and reducing the impact of freezing episodes. This paper explores the implementation of a CNN+RNN+PS machine learning model on a microcontroller-based device. The device operates at a real-time processing rate of 40 Hz and is deployed in practical settings to provide ‘on demand’ vibratory stimulation to patients. This paper examines the system’s ability to operate with minimal latency, achieving an average detection delay of just 261 milliseconds and a freezing of gait detection accuracy of 95.1%. While patients received on-demand stimulation, the system’s effectiveness was assessed by decreasing the average duration of freezing of gait episodes by 45%. These preliminarily results underscore the potential of personalized, real-time feedback systems in enhancing the quality of life and rehabilitation outcomes for patients with movement disorders.

## 1. Introduction

Freezing of gait (FoG) is a common gait disorder characterized by sudden episodes where Parkinson’s disease (PD) patients hardly initiate or continue walking, thus affecting their gait negatively [1,2]. Parkinson’s disease is the second most common neurodegenerative disorder worldwide and affects circa 0.3% of the population around the world, among which 80% of PD individuals eventually experience FoG episodes [3]. The disease is caused by a deficiency of dopamine in the basal ganglia, causing typical motor symptoms such as resting tremor and rigidity, which affect gait motion seriously and lead to higher risk of falls and mobility limitation, thus affecting the life quality of PD patients [4].

As the disease progresses and the gait becomes worse over time, the symptoms are mostly in reduced step/swing length and/or instability, which produces medical and also social challenges [5,6]. The most common medication for PD patients involves dopaminergic drugs to address motor disturbances, however, freezing of gait is often resistant to such treatment and can lead to falls [7,8]. With the unpredictable nature of FoG, there is increasing interest in developing various machine learning-based approaches for real-time FoG detection. Firstly, a good model of human gait must be obtained so gait analysis can reveal FoG episodes. Modeling can be achieved through different approaches such as external visual recordings and image processing, pressure-sensitive tiles, body markings, or a wearable system measuring various sensor information mounted on different body points [9]. Data processing and modeling are primarily performed using machine learning algorithms to achieve higher model accuracy, which in turn improves the detection capabilities and overall robustness of the FoG system [10]. For practical reasons, the FoG-detecting systems are desired in the form of wearable devices, which offer promising means on a daily basis for indoor/outdoor conditions and provide a critical time window for on-demand interventions. Therefore, the design and integration of ML algorithms to wearable devices became very important for achieving robust continuous monitoring and real-time detection of FoG [11,12,13]. One of the latest systematic reviews on wearable devices, such as in Huang et al. [14], included thorough overview of type of sensors used, walking task, mounting point, classifier accuracy, type of PD data (patients or dataset), etc. In the literature overview, most wearable systems or devices typically integrate data from inertial measurement units (IMU) based on accelerometers, gyroscopes, and sometimes magnetometers, which capture gait dynamics, so that ML can identify gait specifics and FoG patterns [15,16]. Our previous work focused primarily on the development and integration of such a wearable FoG detection system, with emphasis on the challenges in optimizing wearable devices for daily use, battery life, and user comfort [17,18]. Such light wearable devices can be placed onto different body places in a manner that does not obstruct the walking of PD patients. The decision for mounting point is debatable, some authors prefer sensors on the foot (shoe) [19,20,21,22], some on the lower part of legs (ankle, bellow knee [23,24]), some stick to hip or lower-back [25], some on the wrist [26], and some use a combination with many sensors to achieve better results [27]. In our case, the chosen mounting point was based on preliminary testing, which gave the best results with sensors placed just under the knee.

The development of electronics and sensors for wearable FoG devices is rapidly advancing. In designing our system, we carefully selected electronics based on factors such as processing power, weight, energy consumption, cost, and dimensions, with the goal of unobtrusive, accurate, and fast FoG detection and cueing devices for PD patients. A comparison of various shank-mounted IMU sensors and processing devices revealed similar demands (consumption, weight) and constraints (CPU, RAM) during development. Some older hardware examples used data processing systems developed on IntelXScale systems and Linux [28], with 6 h autonomy and weighted 231 g. Its Bluetooth wireless IMU sensors mounted on the body points used smaller batteries and weighed 22 g. One of the newer systems [23] uses STM development boards such as the neMEMSi platform, with 9-axis IMU LSM9DS0, a Bluetooth V3.0, and an ultralow-power 32-bit microcontroller STM32L1. In [29], authors used sensors wirelessly connected to a smartphone as wearable device and PC for data storing and processing. Mazzety et al. [30] used prototype Bio2Bit Move by STM with one differential channel for sEMG acquisition ST HM121, an IMU LSM6DS3H, a ARMR CortexR -M4, a microSD, Bluetooth 4.0, and a 592 mWh battery, with a total weight of 10 g including the battery. Rodriguez et al. [31] reviewed some opensource wearable devices with dataloggers and sensors; for instance, the 6-DOF IMU Shield from DFRobot, or FreeIMU, a 9-axis IMU, in combination with an Arduino platform with an Atmega368 microcontroller. Another low-cost option can be Sparkfun’s UDB5 that uses a dsPIC33FJ256GP710, or the x-IMU from IO-Technologies, with a 10-axis IMU and a dsPIC33FJ256GP804. Suppa et al. [32] used an ultralow-power 32-bit microcontroller based on the Cortex™ M3 architecture, with two IMU LSM9DS0 and Bluetooth V3.0 communication, while the offline post-processing was performed by a PC.

For our lightweight, battery-efficient FoG detection and cueing device, we selected market-available components. The FoG detection system used an Arduino Portenta H7 microcontroller with an STM32F747 processor (dual-core: Cortex-M7 at 480 MHz and Cortex-M4 at 240 MHz) and integrated WiFi/BT, programmable in microPython 3.4 or Arduino IDE 2.3.3. The microcontroller is powered by a 1000 mAh Li-Po battery, providing 6 h of autonomy with 550 mW consumption. The device weighs 57 g. The cueing systems use an Attiny 85 microcontroller, HC-12 RF modules, and DC vibrational motors. Our study, conducted on real PD patients, showed excellent FoG detection accuracy. Our classifier algorithm achieved accuracy of 95%, where some authors claim even more [10,33]; however, a lot of reported results were achieved on few test samples, but our experience showed that the most problematic thing for FoG algorithms is the variability in FoG presentations (recorded data) across individuals, therefore forcing the development of personalized detection algorithms that can adapt to a patient’s specific gait patterns [34]. To address this issue, we created an ML algorithm based on the combination of convolutional neural networks (CNNs) and recurrent neural networks (RNNs) with added past samples (PS) to form a more accurate ML algorithm, which was integrated into a wearable device. Similarly, ML algorithms in [35,36] offer a robust partial solution for analyzing the complex gait data collected from wearable sensors. CNNs are used for extracting features from multidimensional sensor data, RNNs capture the temporal dependencies necessary for detecting FoG effectively and PS for the optimization of the individual data to enhance model performance, hence leading to more personalized and adaptive detection systems. The achieved robust FoG episode detection system forwards an alert by means of a cueing system to the PD patients, hopefully preventing their fall or other possible injury.

The integration of a robust wearable feedback system for PD patients is gaining attention since they provide external stimuli to help overcome FoG episodes. Essentially, they divide many types (auditory, visual, and tactile) of cues, each offering unique mechanisms and benefits. The most researched methods for managing FoG are auditory cues, such as rhythmic sounds or metronomes, which help stabilize the gait by synchronizing the gait to a regular rhythm. A meta-analysis by Spaulding et al. [37] demonstrated that auditory cueing improves gait velocity, stride length, and reduces FoG duration in PD patients significantly. Recent innovations integrated auditory cueing into wearable devices, enabling real-time cue delivery. For example, the “AmbuloSonus” system [38] uses foot-worn sensors to detect FoG and deliver rhythmic auditory cues via wireless earphones, which showed a reduction in FoG frequency and severity in clinical trials. The second group of visual cues are delivered mostly through augmented reality glasses or head-mounted displays that provide spatial or temporal references to help individuals navigate FoG episodes. Research indicates that visual cues, such as laser lines projected onto the ground, can enhance gait regularity significantly and reduce FoG occurrence. However, the practical implementation of visual cueing systems faces challenges such as integrating wearable displays into daily life and avoiding sensory overload in complex environments [39,40]. The proposed paper incorporates the third type of cueing by using tactile cues, such as vibrations or mechanical stimuli delivered through wearable devices such as smartwatches or belts, to prompt movement or alter gait patterns during FoG episodes. Tactile cueing is a discreet, non-intrusive intervention, making it suitable for public environments. A study by Caldas et al. [41] showed that wearable devices delivering rhythmic vibrations to the wrist or ankle effectively reduce the FoG episode duration.

The approach presented in the paper is based on alerting patients of an upcoming FoG episode by using a vibrational cueing system. The existing preliminary results of patient stimulation suggest its effectiveness [42,43,44,45]. A review in this field showed many different approaches for stimulating PD patients, such as those mentioned: auditory [42,43,44], visual [46,47,48,49], vibratory [43,50,51,52], electrical stimulations [45,53,54], or a combination of different stimulations [55]. The preliminary results [42,51,53,54,55] suggest that additional stimulation for patients can help reduce the duration of FoG episodes. Continuous stimulation has, thus, become a commonly used approach; however, over time, patients may become accustomed to the stimulation, reducing its effectiveness. To address this issue, the effectiveness of the stimulation could be extended through a system that enables real-time detection of FoG episodes and delivers active stimulation prior to the FoG episode. Despite extensive research in this field, we found an insignificantly small number of studies that investigate specifically the active delivery of vibrational stimulation in response to FoG detection [52].

## 2. Materials and Methods

This chapter presents a comprehensive overview of the materials, methods, and technological innovations employed in the development of a real-time gait analysis system for detecting and addressing freezing of gait (FoG) in Parkinson’s disease (PD) patients [56,57,58]. The three main components described in this chapter are vibrational stimulation actuators, the real-time gait analysis module, and the utilized machine learning algorithms. Additionally, this chapter outlines the methodological framework, including the implementation of the ML algorithm in an online environment and the dataset recording protocol, ensuring ethical standards and patient safety throughout the research process.

### 2.1. Materials

Five vibrational motors were purchased from NFPmotor (Kowloon, Hong Kong). Six 1000 mAh Li-Poly batteries were purchased from HTE d.o.o. (Maribor, Slovenia), and used to power a real-time gait analysis system and five vibrational stimulators. A single-cell Li-Poly 3 A battery management system (BMS) was purchased from Open Electronics U.A.B. (Kėdainiai, Lithuania) and utilized for safe battery charge and discharge cycles. An Arduino Portenta H7 microcontroller was purchased from Farnell (Ljubljana, Slovenia). The HC-12 RF modules were purchased from TinyTronics b.v. (Eindhoven, The Netherlands).

#### 2.1.1. Vibrational Stimulation Actuators

The choice of vibrational stimulators seemed most appropriate, as they operate non-invasively, consequently posing no risk to patients [59]. The universality of the stimulators can be ensured by their wireless operation, which eliminates the need for copper connections between the microcontroller that detects FoG and the stimulators themselves. Five vibrational stimulators were developed, each containing a miniature Attiny 85 microcontroller measuring 23 × 18 mm. This is an 8-bit microprocessor with 8 kB of memory, operating at 20 MHz. The HC-12 modules ensure wireless operation.

A DC vibration motor was chosen that weighs 14 g, consumes 300 mA of current at 3.7 V, and reaches a speed of 4800 rpm. The current consumption of the vibration motor is too high to be powered directly from the microcontroller, so an additional 2N222 transistor activated by the digital output of the microcontroller is used to amplify the current. The actuators are powered by a li-ion battery with a 1000 mAh capacity, where a BMS module is used for battery protection. The housing for the electronics, measuring 69 × 41 × 17 mm, are printed from ABS plastic using an FDM 3D printer. The finished vibrational stimulator weighs only 60 g, making it unobtrusive for the user.

Figure 1 shows the development of five vibrational stimulators, which monitor the RF connection continuously. Upon FoG detection, the stimulators receive RF command and begin to vibrate rhythmically at a rate of 1 Hz (approximate walking speed). The power of the vibration is adjustable via RF commands, allowing a choice between ten different levels of vibration intensity. All five vibrational stimulators can be activated simultaneously through the RF module using a single RF command.

#### 2.1.2. Real-Time Gait Analysis Module

The machine learning models used in our previous paper [17] were initially trained and tested in an offline environment using a PC. The next step is the implementation of the trained ML model onto a microcontroller to enable its online execution.

A data capture and storage module equipped with a more powerful microcontroller was developed to enable real-time classification. This upgraded microcontroller enables both data capture and storage, as well as simultaneous real-time gait classification, as it has the trained ML model implemented on it.

A miniature Arduino Portenta H7 microcontroller (measuring 66 × 25 mm) was selected, which incorporates an STM32F747 processor (8 Mb SDRAM, 16 Mb NOR Flash) with two cores: ARM Cortex-M7 (480 MHz) and Cortex-M4 (240 MHz). Each core can run its own program concurrently and the cores are capable of intercommunication. The Arduino Portenta H7 also includes an integrated WiFi and BT module and can be programmed in the microPython or Arduino IDE environments.

A dedicated printed circuit board (PCB) was developed that adds an HC-12 RF module, a micro-SD module, a power switch, a BMS for battery protection, an additional fuse, and an I2C connector compactly for connecting the sensors to the microcontroller. This expansion PCB connects directly to the Portenta H7 microcontroller, maintaining the compactness of the overall gait analysis system. Figure 2 illustrates the development of the module for real-time gait analysis.

A measuring system containing accelerometers, gyroscopes, and muscle activity sensors integrated into an elastic strip [17] is utilized to periodically measure user’s gait. A micro-electromechanical system (MEMS) based accelerometers and gyroscopes contains a certain amount of noise [60], which can be addressed using a complementary filter [61].

A housing measuring 70 × 30 × 24 mm was made using an FDM 3D printer. A 1000 mAh Li-Po battery is used, which provides 6 h of autonomy with a module consumption of 550 mW (150 mA at 3.7 V). The finished module for real-time gait analysis weighs only 57 g. Upon FoG detection, the real-time gait analysis module sends an RF command that activates the vibration stimulators which begin delivering cues to the PD patient.

Figure 2c also shows two additional contacts. These are digital or analog outputs, which can be used to activate any stimulator upon FoG detection. A speaker (buzzer) can also be installed on this output to stimulate the patient audibly (such as a metronome).

### 2.2. Methods

This chapter outlines the methodologies employed in the development of a real-time gait analysis system, focusing on the selection of machine learning algorithms and the challenges encountered throughout the project. For clarity and thorough exploration, the chapter is organized into three main sections:machine learning algorithm,implementation of the ML algorithm in an online environment, andthe dataset recording protocol.

#### 2.2.1. Machine Learning Algorithm

The quality of ML algorithms depends primarily on the information used for their training (the quality of input determines the quality of output). The input data must contain enough information that includes some form of meaning (and not random information, i.e., noise), so that the ML model can learn the desired task successfully [62].

The robustness and effectiveness of combining a convolutional neural network with recurrent neural networks for classification of different gait activities was demonstrated in our previous paper [17], where we compared nine different ML algorithms, as well as in the reviewed literature [63,64,65,66,67,68]. For online implementation of the ML algorithm in the C programming language, we need to consider additional factors, such as the computational and implementation complexity of the algorithm itself. In a real-time environment, we want to avoid computationally intensive operations such as continuous wavelet transform (CWT) if we want to achieve an analysis speed of 40 Hz.

We decided on a CNN+RNN ML algorithm architecture that can process current sample and past samples (PS) simultaneously. The CNN+RNN+PS ML algorithm is trained and tested on preprocessed gait data that can include a maximum of 100 past samples. The number of past samples incorporated during training is treated as a hyperparameter, which is optimized systematically for each individual to enhance model performance.

The preprocessing of training data by adding past samples is computationally non-demanding and suitable for implementation on a microcontroller, while also allowing the ML model to process a broader temporal data window [69]. The CNN+RNN+PS ML algorithm with added past samples, as shown in Figure 3, receives data in the format of 800 × 1, representing the latest sample followed by 39 past samples. In this way, the ML model receives a wider data window for each new sample that we wish to classify. In the offline preprocessing stage, 800 input features are generated from 20 input signals for each sample in the dataset. The online past samples preprocessing is only a matter of creating and maintaining the rolling buffer. It is essential to ensure that the preprocessing in both online and offline environments are identical.

By utilizing past samples, machine learning models can gain a deeper understanding of motion data, as the model does not only process a single sample, but analyzes multiple past samples simultaneously, allowing for more accurate classification of the current sample. This method increases the data dimension without requiring additional operations, making it suitable for real-time implementation. The selection of the number of past samples can be an iteratively optimized process tailored to each individual user. This ensures an additional level of personalization for ML algorithms for each person. Equation (1) defines the input vector for the CNN+RNN+PS ML algorithm:(1)$$v={\left[u_{1,t},\;{\;u}_{2,t},\;{\;u}_{3,t},\ldots ,\;{\;u}_{20,t},\;{\;u}_{1,\;t-1},\;{\;u}_{2,\;t-1},\;\ldots ,\;{\;u}_{20,t-1},\;\ldots \ldots ,\;{\;u}_{20,\;t-PS}\;\right]}_{,}$$
where v represents a 1-dimensional vector including past samples, $u_{i,t}$ represents the measurement of the i-th sensor at time t, where i is the sensor index (from 1 to 20), t is the current sample, t−1 is the previous sample, and t−PS is the last sample (*PS* represents the number of past samples included).

The CNN+RNN+PS ML model first uses a reshaping layer to convert the 800 input features back into the format of 20 × 40, which are then processed by the CNN layers. Using a flattening layer, the data are converted into a format of 1280 × 1, which is processed by two RNN layers. The number of past samples is optimized iteratively as a hyper-parameter with the goal of quality classification [70]. The pseudocode and the configuration for the CNN+RNN+PS ML algorithm is included in the Appendix A.

The CNN+RNN+PS ML algorithm in Figure 3 has five output neurons meant for classification of five gait activities. For FoG detection, only 1 neuron is sufficient.

Due to the exponential dependency on the number of inputs and the size of the model, the CNN+RNN+PS ML algorithm is not suitable for classifying CWT data with additional past samples. Subsequent results for the CNN+RNN+PS ML model, trained with CWT data, pertain to the use of only the current (single) sample.

#### 2.2.2. Implementation of the ML Algorithm in an Online Environment

The Matlab software environment served as a solid foundation for learning and testing offline ML models. The Arduino Portenta H7 microcontroller is programmed in the C language. Initial attempts to transfer ML models from the Matlab environment to C code were not successful. The Python environment with the TensorFlow ML library proved to be more suitable for learning, converting, and implementing ML algorithms into C code.

Manual normalization of gait data, as presented in our previous paper [17], proved inefficient for ML algorithms implemented on the microcontroller, as it requires real-time normalization of the data. Normalization is carried out according to Equation (2):(2)$$y_{t}=\frac{x_{t}-µ}{\sigma },$$
where $x_{t}$ is the sample being normalized, *μ* is the average among the data, and *σ* is the standard deviation. In the offline mode of training and testing ML models, the entire dataset is known beforehand, which means that *μ* and *σ* can be calculated for the entire dataset. In the online implementation of the algorithm, we have the trained ML model implemented on the microcontroller, and now we test it in real time. Upon the arrival of new samples for classification, it is necessary to calculate *μ* and *σ* using Equation (1). Since we only have a brief snippet of gait data, we cannot determine those variables accurately. A temporary solution involves calculating variables for the entire training dataset, which we then transfer to the microcontroller. Although this method suffices temporarily, as the dataset grows over time, the calculated variables may no longer be relevant for a new dataset, and the ML model becomes less accurate with time. Therefore, a more robust method is required, allowing for more flexible and dynamic data normalization.

It turns out that a single normalization layer within the CNN+RNN+PS ML model operates dynamically and is all we need for effective data normalization. In this way, normalization is performed within the model automatically. The normalization layer has two parameters, i.e., beta and gamma, where beta represents the learned scaling factor and gamma the learned offset factor. During the learning process of the model, these parameters are optimized with the goal of improving the classification quality. During the testing of the model, the normalization layer uses the learned beta and gamma parameters for dynamic normalization of new samples (beta and gamma are not determined statically).

The CNN+RNN+PS ML model was proven to be suitable for implementation on a microcontroller, considering the quality of classification and the computational efficiency of the algorithm. The implementation of the CNN+RNN+PS ML algorithm in the Python environment is presented next.

Python is a dynamic high-level, interpreted language, known for its simplicity and code readability. TensorFlow (TF) is an open-source library for the Python environment, used for developing and learning ML models. Due to its adaptability and the ability to operate on various platforms, TF enables easier implementation of complex algorithms. A Python 3.9 environment is created, with the following libraries installed: TF 2.5, TFGPU 2.5, KERAS TUNER 1.3.0, and NUMPY 1.23.1.

This combination of libraries allows for the learning of ML models using a graphics card, which speeds up the execution of algorithms about six times, thereby enabling faster development and evaluation of ML models.

Using the TF library, the CNN+RNN+PS ML algorithm was implemented in the Python environment, with the goal of achieving the same quality of classification as its predecessor in the Matlab environment.

TensorFlow Lite is a lightweight version of the TF library, designed specifically for mobile and embedded devices. It enables fast and efficient execution of ML models on devices with limited resources. TensorFlow Lite achieves this by optimizing ML models for fast execution, using less memory.

The trained CNN+RNN+PS ML model in the TF environment can be converted into a model.tflite file. The model.tflite file can then be visualized and analyzed using the netron.app tool, available online. In the next step, the model.tflite file is converted into hexadecimal (Hex) values. The ML model, written with hex values, contains both the structure and the learned weights and biases of the model.

The pseudocode for performing both model conversions is included in the Appendix A of the paper.

In the Arduino environment, we can use the model (written in a hex file) to perform classification using the TFlite library. The Arduino Portenta H7 microcontroller contains two cores, which can execute different tasks simultaneously. The M7 core is programmed with the trained CNN+RNN+PS ML model, while the M4 core is programmed with a program that captures and stores the gait data at a rate of 40 Hz. With each new data sample arrival, (every 25 ms), through a remote procedure call (RPC) communication, the M4 core sends the data of the new sample internally to the M7 core. The M7 core preprocesses the new data with past samples and forwards the data to the ML model, runs the model, and stores the model’s calculated output. If an FoG is detected for a current sample, an RF command is sent to the vibrational stimulators, which begin cueing with minimal latency. The system is configured to provide cues for 3 s from the last detected FoG sample, where the cue duration is adjustable.

It is important that the ML model in the Python and Arduino environments are treated identically. Arduino uses a float type of variable, where values are stored using 32 bits (4 bytes), while TensorFlow, by default, uses a float 64 type of variable. Thus, a crucial step is to train the ML model in the Python environment with 32-bit float variables (which is achieved by initializing the training variables as 32-bit floats).

It is also important that the data preprocessing process in the Python environment is identical to the data preprocessing in the Arduino environment. On the Portenta microcontroller, preprocessing is implemented as a rolling buffer, where, with each new sample, the memory shifts one position to the left, and the new (latest) sample is added to the buffer. This results in a matrix of size 800 × 1, which is updated in real time. Figure 4 below illustrates the flow diagram of the CNN+RNN+PS ML algorithm in both offline and online execution environments.

#### 2.2.3. Participants and the Dataset Recording Protocol

All the participants in the study were informed of the purpose and process of the measurements. Before beginning their participation, they provided their written statement of conscious and voluntary consent to the research. The execution of the measurements was conducted in accordance with the Helsinki Declaration, which establishes ethical standards for biomedical research on humans, as well as the provisions of the Council of Europe’s Oviedo Convention, which protects human rights and dignity in biomedicine. We also adhered to the European Code of Conduct for Research Integrity, additional protocols to biomedical research (CETS No. 195), relevant legislation on ethics in research, and the principles of the Slovenian Code of Medical Deontology, thereby ensuring the highest standards of ethics and integrity in our research work. For the measurement of patients, we obtained approval and a Certificate of Ethical Acceptability of the research from the Slovenian Commission for Medical Work (No. 0120-46/2024-2711-5). The measurements of patients were conducted at the Neurological Clinic in Ljubljana (Slovenia).

A 10 to 15 min recording of the patient’s movement is typically sufficient for training the ML model, with data diversity being key. The recording should capture various daily activities such as resting, walking, rotating, walking through doors, and FoG episodes. Since FoG is the focus, the recording must include as many freezing episodes as possible. FoG episodes are unpredictable, influenced by physical and psychological factors, with patients often experiencing motor slowness, tremors, and fatigue due to the neurodegenerative nature of the disease. This results in lower quality, limited gait data with unrecorded boundary conditions, and unbalanced distribution. To ensure safety and prevent falls, two trained individuals monitored the subject, with sharp objects and obstacles removed. A movement polygon was created in the lab for the patient to perform everyday tasks such as walking, turning, and sitting. Occasionally, they were asked to count numbers to increase cognitive load and induce FoG episodes. Table 1 shows the dataset distribution among nine PD patients.

Table 1 reveals a significant imbalance in the recorded datasets activity distribution, potentially causing classification issues [71]. Person No. 2’s dataset is the most unbalanced, with FoG episodes making up only 1.5% of the data. The first six patients’ movement data were used for offline FoG detection, while real-time detection was tested on the seventh, eighth, and ninth patients. The first six were recorded using a gait acquisition module from a previous paper [17], while the last three were recorded after the online gait analysis module was completed. Section 3.1 and Section 3.3 detail offline and online FoG detection, respectively.

## 3. Results

Our primary objective is focused on the offline detection of freezing episodes, evaluating the accuracy, reliability, and selectiveness of detection. Subsequently, we performed an online detection test on a healthy subject to demonstrate that the system activates vibratory stimulation selectively only when a freezing of gait is simulated. In the final phase, the online FoG detection system was tested in real time on actual patients with PD, where we preliminarily demonstrated and subsequently evaluated its effectiveness. Quality assessment metrics are essential, not only during the ML model training process, but also in its validation and when comparing different ML models against each other [72].

### 3.1. Offline Detection of Freezing of Gait Episodes

In our first paper [17], we classified extensive and rich gait data successfully into five different activities, where this data contain plenty of relevant information and a minimal amount of noise. However, the measurement protocol for actual patients differs significantly from the previously tested protocol. This is reflected in the collected dataset, which is often more limited due to the specificity of the measurements, and sometimes does not include boundary conditions. Table 1 presents the distribution of activities, where FoG represented only an average of 4.5% of samples among all the collected data.

The purpose of the experiment was to evaluate the CNN+RNN+PS ML model for offline detection (classification) of FoG in six patients. The CNN+RNN+PS ML model is chosen because it is simple, efficient, and accurate in classifying raw datasets. FoG detection is a binary classification problem, where data are labeled with 1 (FoG) and 0 (all other activities). Targets are used as the desired output of the model in evaluating classification, where the output of the model and the data label (targets) are compared, and evaluation metrics are calculated. The CNN+RNN+PS ML model is tested on raw and CWT data. For each subject, 60% of the movement data are used for training the ML model, and the remaining 40% for testing the model. The results obtained with the CNN+RNN+PS ML model for the detection of FoG for the first six patients are presented in Table 2.

From Table 2, it is evident that there is a high rating of accuracy and specificity, while the ratings for precision and sensitivity are extremely low, both for raw and CWT data. It is important to mention the strong imbalances in input data, as shown in Table 1. Imbalances in the data are reflected in the imbalances of calculated true-positive (TP), true-negative (TN), false-positive (FP), and false-negative (FN) variables, which, subsequently, affects the calculations of the evaluation metrics. This problem could be addressed by using an adjusted calculation of metrics that accounts for data imbalances with weights.

CWT is a computationally demanding method, and we aimed to avoid it when running the algorithm in real time. A raw dataset of gait data was used for all following tests, validations, and visualizations of ML models.

The performance of trained models can also be checked through visualization, where we display the output of the model and targets. We were interested in whether the model reacts appropriately in identifying FoG. Visual analysis of the output allows a deeper insight into the behavior (response) of the model and confirms its effectiveness in practice. Figure 5, Figure 6, Figure 7, Figure 8, Figure 9 and Figure 10 present the outputs of the CNN+RNN+PS ML model, trained with a raw dataset of gait data visually, for six patients.

From the above six figures it is evident that the CNN+RNN+PS ML model generally succeeds in detecting all marked FoG events. However, the output from the model during FoG detection was not constant, often oscillating between values 0 and 1. For implementing FoG detection in real time, this does not pose a problem, as upon the first detection of FoG, when the model first returns a value of 1, stimulation is initiated, which lasts until the model records at least 120 consecutive samples (3 s) without a detected FoG (output value 0). The average time delay for detecting freezing of gait was 10.45 samples, which amounts to only 261 ms. In the above figures we observe some examples of false-positive (FP) FoG detections, which highlights that marking FoG events accurately during the recording of gait data is a complex challenge.

Despite expert knowledge, sometimes we only guess whether a patient truly experienced FoG. The labeling of FoG events is performed based on visual confirmation by two trained observers and there remains a possibility that some less pronounced FoG events were present and were not marked correctly. An example supporting this hypothesis is the Figure 5b graph, where FoG was detected with an uncertain decision by the model, while an example of FP detection at the 9500th sample shows high certainty of the model in detecting FoG. It is not necessarily bad that stimulation is still triggered in cases of FP detections, as there is a certain similarity in the data between FP cases and correctly marked FoG events (this is reflected in the model triggering in these cases). Stimulation during FP cases can, nevertheless, have a positive impact on improving the patient’s motor skills.

Figure 5 also confirms the hypothesis that the metric for classification accuracy does not tell the whole story. For the first patient, a classification accuracy of 94.1% was measured due primarily to a high number of TN detection cases.

All the model outputs shown in the Graph (d) in Figure 5, Figure 6, Figure 7, Figure 8, Figure 9 and Figure 10 were processed further with a threshold function. The threshold function works by comparing the model output with a pre-set threshold, and determining the final result based on this. The threshold can be adjusted; in the experiments described, it was set at 0.9. This means that when the ML model’s output exceeded the value of 0.9, FoG was detected, otherwise, it was not detected.

Subsequently, Table 3 presents the evaluation of the CNN+RNN+PS ML model for different combinations of raw sensor training data, consisting of complementary filter (CF), accelerometer (ACC), gyroscope (GYRO), and muscle activity sensors (MA):

From the data presented in Table 3, it is evident that all the data combinations enabled successful classification. However, visualization of the results shows that combinations including three or all four data metrics create an ML model with an output that is cleaner and contains less noise. It can be seen from Table 3 that the F1 score improves gradually with an increase in the amount of data used. It is also noticeable that the inclusion of MA data did not contribute significantly to improving the quality of the ML model. The highest F1 score was achieved by the ML model trained using CF, ACC, and GYRO data.

### 3.2. Online Detection of Simulated Freezing of Gait Episodes

After the successful implementation of the trained CNN+RNN+PS ML model in the Arduino environment, we continued with initial real-time system testing. This phase began with an experiment, where a healthy subject simulated FoG events. This individual simulated FoG despite not having any health issues. This recording included various types of activities: indoor walking, outdoor walking, running, jumping, standing still, sitting, and simulating FoG.

Simulating FoG presents a unique challenge, as it is difficult to reproduce the actual conditions experienced by patients accurately. Nonetheless, this simulation served as an initial validation of the system’s functionality, allowing us to test the responsiveness and effectiveness of the solution designed for recognizing FoG events.

The experiment was designed such that we first trained the CNN+RNN+PS ML model in the Python environment on simulated FoG data, then converted and implemented the trained model into the Arduino C code. In the next step, we recorded a test dataset with the real-time gait analysis system and performed classification in parallel, where all the variables (movement data + ML model output) were also stored on an SD card. The captured test dataset was then transferred back to the Python environment, where it was also used for offline classification. The results of the experiment are displayed in Figure 11.

After reviewing the results from Figure 11, we can observe that all the simulated FoG episodes were detected successfully. We noted that the offline and online classifications were very similar, except for certain minimal differences, which can be attributed to data normalization. Figure 11c,d shows the output of the offline and online CNN+RNN+PS ML model without using a threshold function, while (e) displays the FoG_condition variable, which is used to activate vibratory cues. The average delay between the simulated freezing of gait and the onset of vibratory stimulation was only 9.66 samples (240 ms).

At the beginning of the program, the FoG_condition variable was set to a value of 0. When the output from the ML model exceeds a certain threshold (usually set at 0.9), the FoG_condition variable is updated to a value of 3. When FoG is no longer detected, the value of FoG_condition is reduced gradually by one unit every second (or every 40 samples), allowing the stimulation to continue for three seconds after the last detected FoG episode. The system provides rhythmic vibratory stimulation continuously throughout the period when the FoG_condition variable exceeds 0.

This experiment was successful. The user of the real-time gait analysis system received rhythmic vibratory stimulation only at moments when FoG was simulated (on-demand). The module for real-time gait analysis allows for the setting of the FoG threshold and intensity of stimulation. Adjustments to these variables can be made during real-time system operation using RF commands.

### 3.3. Online Detection of Freezing of Gait Episodes

Following the successful initial experiment of the system’s functionality, validation was carried out in real time on actual patients with Parkinson’s disease. The system was validated on the seventh, eighth, and ninth patients, where the distribution of training data is shown in Table 4. All the individuals experienced freezing of gait during the measurements. However, the amount of data collected for training was quite limited, amounting to only 237 s for the seventh patient, 210 s for the eighth patient, and 351 s for the ninth patient.

With the captured training data, a personalized CNN+RNN+PS ML model was prepared for each individual patient, implemented on the Portenta H7 microcontroller and tested in real time. During the subsequent real-time measurements, the patient was equipped with actuators for delivering vibratory stimulation, mounted on the left and right arms and around the waist. Figure 12 displays the output of the CNN+RNN+PS ML model visually, which detected FoG episodes in real time.

Figure 12 shows that the CNN+RNN+PS ML model successfully detected all eight marked FoG episodes, but there were three false positives. When the model’s output exceeds 0.2, the FoG_condition variable is set to 3, triggering rhythmic vibratory stimulation, which occurred for all eight episodes. For the seventh patient, the model was trained with data including 60 past samples and seven marked FoG episodes. For the eighth patient, despite only four FoG episodes available for training, the model accurately detected all three marked episodes during testing. Figure 13 displays the real-time detection results for the eighth patient.

Figure 13 shows the successful detection of all three marked FoG episodes. We can observe four samples of false-positive FoG detection. The CNN+RNN+PS ML model used movement data containing 100 past samples for real-time detection of FoG episodes. For the eighth patient, the FoG detection threshold was set at 0.3. The CNN+RNN+PS ML model operated very robustly in real time, considering that it was trained and personalized with only a modest set of data containing just four marked freezing of gait samples.

Figure 14 illustrates real-time detection of FoG episodes for the ninth patient, where chart (a) shows the test dataset, (b) shows the target data, (c) displays the real-time output of the CNN+RNN+PS ML model, while (d) shows the FoG_condition variable (periods when the patient received vibrational stimulation).

As shown in Figure 14, only one example of FP detection of FoG was identified for the ninth patient with Parkinson’s disease, while the model detected all six marked FoG episodes successfully. The CNN+RNN+PS ML model used 60 past samples for classification in the ninth patient, with the FoG detection threshold set at 0.4. For the ninth patient, the CNN+RNN+PS ML model had 13 marked FoG episodes available for training.

We note that the CNN+RNN+PS ML model operates robustly and selectively in real time, detecting all FoG episodes for all three patients. PD patients often experience FoG episodes during turning or at the start of walking. Due to the limited or unexplored dataset, a non-selective ML model could potentially detect all turning movements or all starts of walking as FoG episodes. Visual inspection of the FP cases of FoG detection indicates that these never occur during normal walking and are not present at every turn or start of the patient’s walking, which proves the learned selectivity of the ML model.

### 3.4. Preliminary Statistical Analysis

The paper continues with the use of statistical analysis and Z-tests to quantify the preliminary impact of vibratory stimulation. The analysis is based on a comparison of data targets obtained during initial measurements without vibratory stimulation for the purpose of training the machine learning model and data targets collected during real-time system testing, where the patient was equipped with three vibratory stimulators mounted on the left and right arms and torso. Graph (d) in Figure 12, Figure 13 and Figure 14 illustrates the periods when the patient received rhythmic vibratory stimulation. Table 4 quantifies the impact of vibratory stimulation.

The FoG probability in Table 4 represents the average likelihood that a new sample is an FoG episode. It is calculated by dividing the number of FoG samples by the total number of samples within the movement data. The Z-statistic is calculated by normalizing the difference between FoG probabilities in data with and without stimulation, using the standard error (SE) of this difference for normalization, as shown in Equations (3) and (4):(3)$$SE=\sqrt{\frac{r_{1}\times (1-r_{1})}{n_{1}}+\frac{r_{2}\times \left(1-r_{2}\right)}{n_{2}}},$$
(4)$$Z=\frac{r_{1}-r_{2}}{SE},$$
where $r_{1}$ represents the FoG probability without stimulation, $r_{2}$ is the FoG probability with stimulation present, SE denotes the standard error, $n_{1}\;$ is the total number of recorded samples without stimulation, and $n_{2}$ is the total number of recorded samples with stimulation. The *p*-value was calculated using the cumulative distribution function (CDF) of the standard normal distribution.

By comparing the FoG probabilities with and without stimulation, we can observe a 32% reduction in freezing durations for the seventh patient, 44% reduction for the eighth patient, and 58% reduction in freezing durations for ninth patient. For all patients, the Z-score was negative, indicating a reduction in the average duration of freezing episodes. The *p*-value was calculated using the Z-statistics derived from comparing the rates of freezing of gait (FoG probability) between the two conditions (with and without stimulation). The extremely low *p*-value for all patients suggests strongly that rhythmic vibratory stimulation had an effect in reducing the duration and frequency of FoG episodes.

## 4. Discussion

This study preliminarily confirms the feasibility of ML models in detecting specific gait disorders in offline and online environments. This paper extends our previous research on gait analysis by exploring the effectiveness of machine learning methods in detecting freezing of gait episodes further. While our initial work [17] confirmed the feasibility of our system in recognizing specific gait activities, here, we focus on advanced aspects of personalization and real-time algorithm deployment, crucial for the delivery of cues on-demand.

The final experiment for real-time FoG episode detection was conducted on three patients with Parkinson’s disease. Initially, a training dataset was recorded for each patient, used to create a personalized CNN+RNN+PS ML model, which was then implemented on a microcontroller, and further, tested for real-time FoG episode detection. The personalized CNN+RNN+PS model identified reliably all eight labeled FoG episodes for the seventh patient, all three episodes for the eighth patient, and all six episodes for the ninth patient with PD. When visualizing the movement data obtained in the mentioned experiment, a few samples of false-positive FoG episode detections can be observed for all patients. The ML model operates selectively, as it does not trigger with every patient rotation or at the start of every walking session. Upon detection of an FoG episode, rhythmic vibrational stimulation was triggered with an average delay of 261 ms. Due to the consistent output of the CNN+RNN+PS ML model, it is highly likely that the false-positive detections corresponded to minimal step freezes that were not observed and labeled by the data labeling module operator.

Our preliminary findings suggest that the integration of inertial measurement units alone can classify human gait activities effectively, an insight that streamlines the sensor setup and reduces the system’s complexity and cost. This simplification is critical for scaling the application in clinical settings, where ease of use and cost-effectiveness are paramount. Moreover, this study confirmed the robustness of the CNN+RNN+PS ML model, not only in detecting FoG episodes, but also in its capacity to be trained with relatively small datasets while maintaining high accuracy. This model was implemented effectively in a wearable real-time detection system using a microcontroller, demonstrating the potential for real-world applications.

The proof-of-concept use of rhythmic vibrational stimulation showed an average reduction in the duration of FoG episodes by 45%. Preliminary statistical analysis of these interventions suggests significant potential for not only detecting, but also managing FoG preemptively through timely cues. This finding could be transformative, offering a non-pharmacological intervention that enhances patient mobility and quality of life. However, for accurately confirming systems influence in reducing FoG episode duration, a double-blind, placebo-controlled study is required.

Despite expert knowledge, it is very difficult to determine accurately if a patient truly experienced an FoG episode. The CNN+RNN+PS ML model seems to exhibit selectiveness during false-positive detection samples, suggesting unlabeled FoG samples. Nevertheless, stimulating patients during false-positive FoG cases can be beneficial, as the data from these samples resemble that of actual FoG events closely.

## 5. Conclusions

This paper presents a real-time wearable system designed to detect and mitigate freezing of gait (FoG) episodes in patients with Parkinson’s disease, utilizing a combination of convolutional and recurrent neural networks enhanced with past sample data pre-processing. Our approach emphasizes personalized feedback and fast intervention through a microcontroller-based device that operates at a processing rate of 40 Hz, achieving a detection delay of 261 milliseconds. The implementation of our CNN+RNN+PS model yielded detection accuracy of 95.1% and preliminary reduction in FoG episode duration by 45%. These findings highlight the potential for this system to serve as an effective non-pharmacological intervention, enhancing mobility and quality of life for patients. While the preliminary results are promising, further validation through double-blind, placebo-controlled studies is essential to comprehensively assess the system’s efficacy and confirm its potential in clinical practice.

## Figures and Tables

**Figure 1 bioengineering-11-01048-f001:**
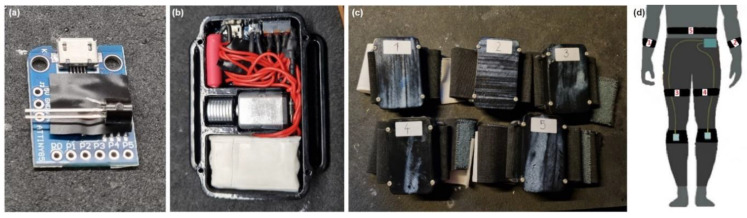
Actuators for vibration stimulation. (**a**) Shows the Attiny85 microcontroller and 2N222 transistor; (**b**) displays the interior of the vibration stimulator; (**c**) shows the completed set of 5 vibration stimulators mounted on elastic bands; and (**d**) a sketch of the potential placement of the vibration stimulators on a patient.

**Figure 2 bioengineering-11-01048-f002:**
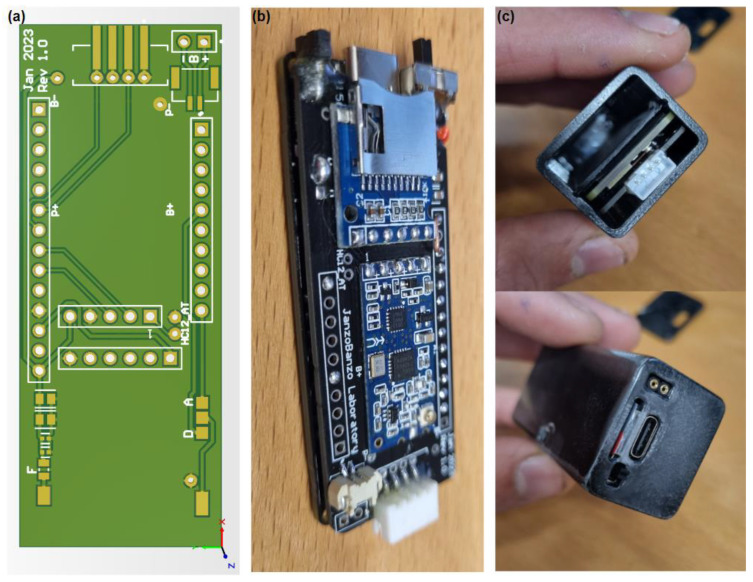
Module for real-time gait analysis. (**a**) Shows the 3D model of the printed circuit board; (**b**) displays the finished PCB installed on the Arduino Portenta H7 microcontroller; and (**c**) illustrates the completed module for real-time gait analysis.

**Figure 3 bioengineering-11-01048-f003:**
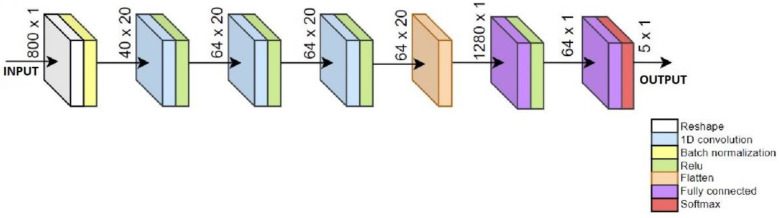
Sketch of CNN+RNN+PS ML algorithm using 40 past samples.

**Figure 4 bioengineering-11-01048-f004:**
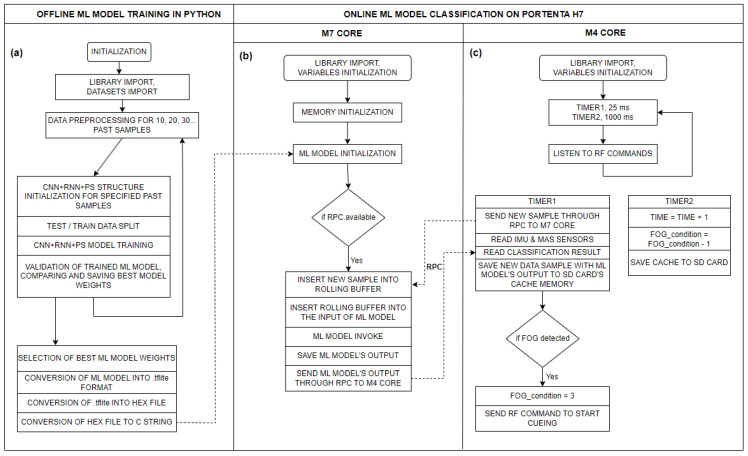
Flow diagram of the CNN+RNN+PS ML algorithm for: (**a**) offline execution of the ML model on a PC in a Python environment (training the model); (**b**) real-time execution of the ML model in C code on the M7 core of the Portenta H7 microcontroller (using the trained model); and (**c**) flow diagram of the measurement system on the M4 core of the Portenta H7 microcontroller.

**Figure 5 bioengineering-11-01048-f005:**
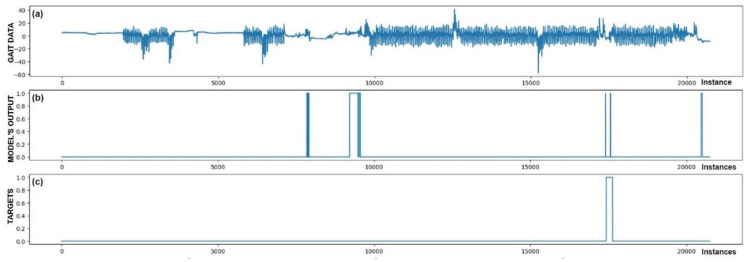
Detection of FoG for the first Parkinson’s disease patient. (**a**) The graph shows the test dataset; (**b**) displays the output of the CNN+RNN+PS ML model trained with raw data; and (**c**) shows the labeled targets. The marked FoG at sample 17,500 was detected successfully. It is observed that the detection was not constant, but the model was triggered twice for a short duration. Three FP cases of FoG detection are observed. The strong FoG detection at sample 9500 suggests that the patient’s freezing of gait episode may not have been labeled during data collection.

**Figure 6 bioengineering-11-01048-f006:**
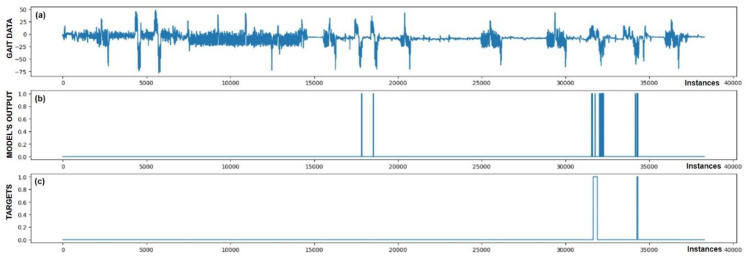
Detection of FoG for the second Parkinson’s disease patient. Both marked FoG events were detected successfully. Again, the detection of FoG was not constant; the model’s output oscillated between 1 and 0 during the actual FoG presence. Two FP detections of FoG are observed at samples 18,000 and 19,000.

**Figure 7 bioengineering-11-01048-f007:**
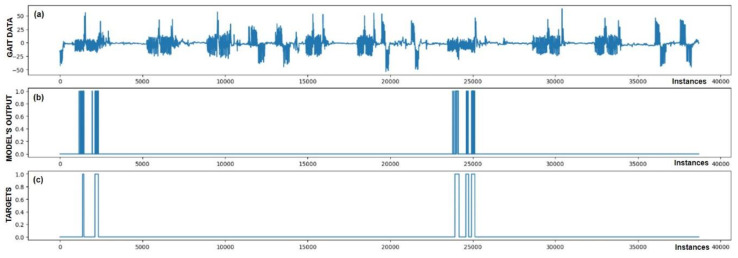
Detection of FoG for the third Parkinson’s disease patient. All five marked FoG events were detected successfully. No FP detections of FoG were recorded.

**Figure 8 bioengineering-11-01048-f008:**
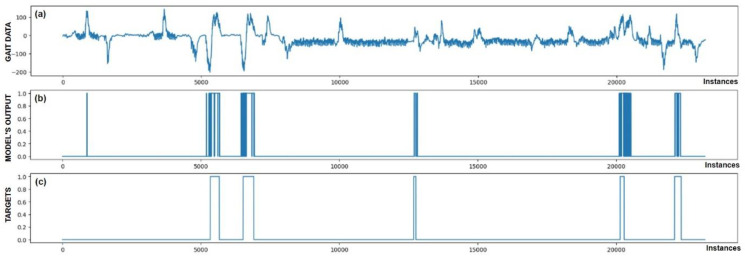
Detection of FoG for the fourth Parkinson’s disease patient. All five marked FoG events were detected successfully. At sample 800, one FP detection of FoG can be observed.

**Figure 9 bioengineering-11-01048-f009:**
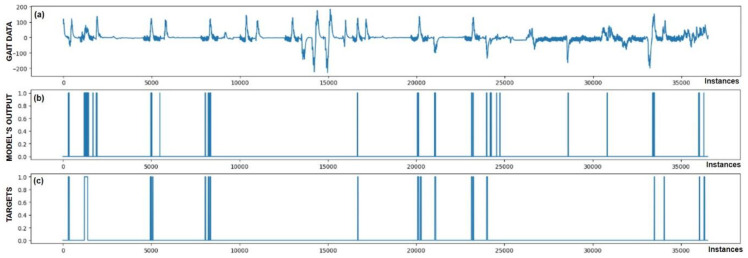
Detection of FoG for the fifth Parkinson’s disease patient, where FoG episodes were detected mostly correctly. However, six samples of FP detections and one FN detection at sample 34,000 were observed.

**Figure 10 bioengineering-11-01048-f010:**
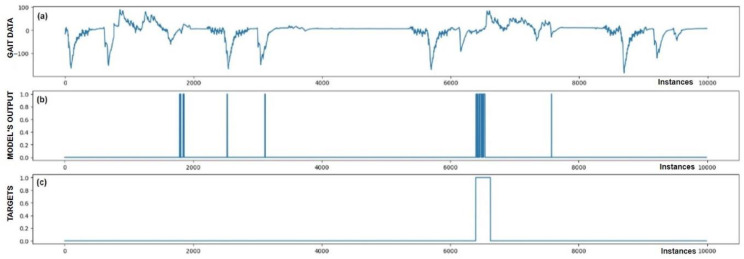
Detection of FoG for the sixth Parkinson’s disease patient. In addition to the successfully detected FoG at sample 6500, five FP detections of FoG were also observed.

**Figure 11 bioengineering-11-01048-f011:**
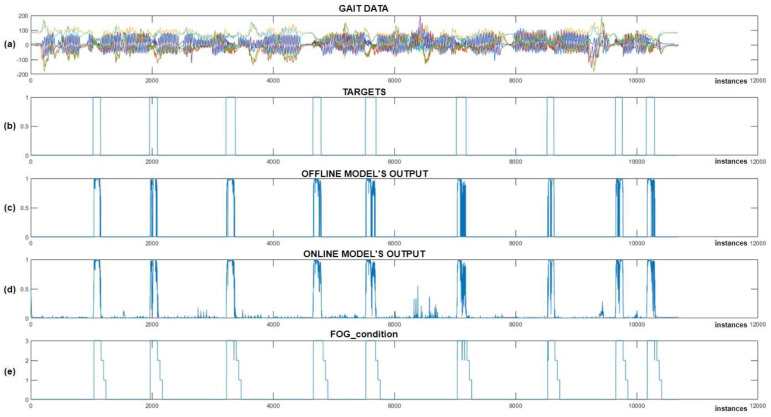
Online vs. offline detection of simulated FoG episodes. (**a**) Represents the test dataset; (**b**) represents the data targets; (**c**) Shows the output of the CNN+RNN+PS ML model in an offline environment; (**d**) shows the output of the CNN+RNN+PS ML model in an online environment; and (**e**) illustrates the ‘FoG_condition’ variable, which activates vibration stimulation.

**Figure 12 bioengineering-11-01048-f012:**
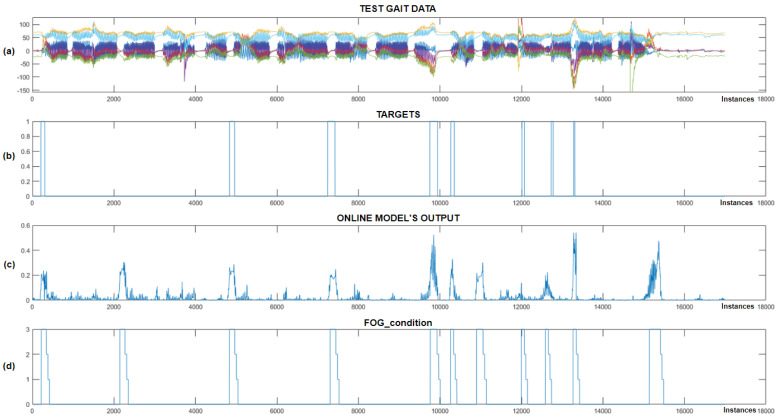
Real-time detection of FoG episodes for the seventh Parkinson’s disease patient. (**a**) Displays the test dataset; (**b**) shows the corresponding targets; (**c**) presents the output of the CNN+RNN+PS ML model in real time; and (**d**) illustrates the FoG_condition variable that activates the vibration stimulation.

**Figure 13 bioengineering-11-01048-f013:**
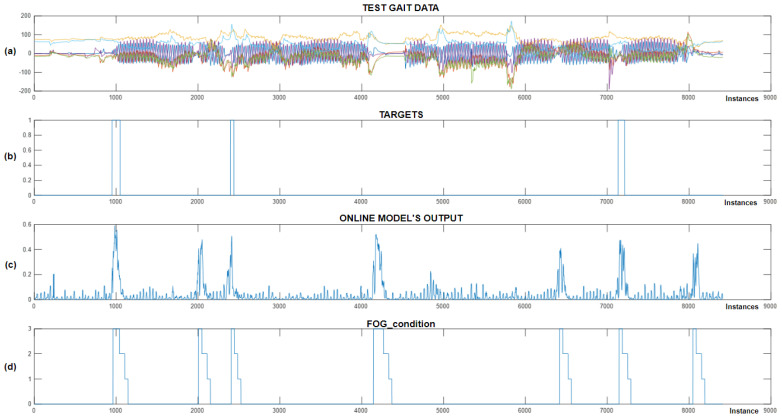
Real-time detection of FoG episodes for the eighth Parkinson’s disease patient. (**a**) Displays the test dataset; (**b**) shows the corresponding targets; (**c**) presents the output of the CNN+RNN+PS ML model in real time; and (**d**) illustrates the FoG_condition variable.

**Figure 14 bioengineering-11-01048-f014:**
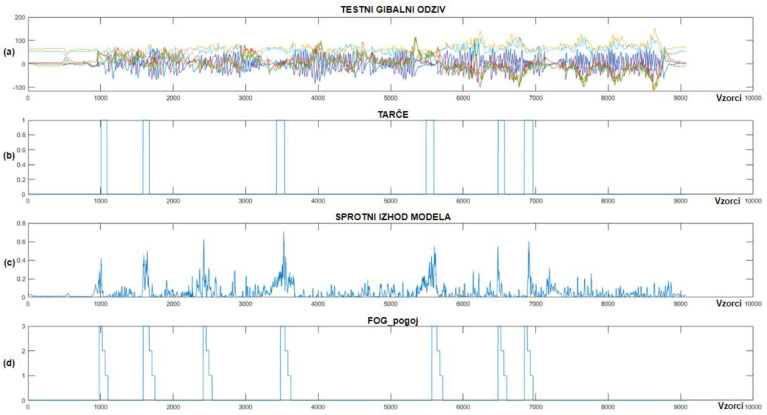
Real-time detection of FoG episodes for the ninth Parkinson’s disease patient. (**a**) Shows the test dataset; (**b**) displays the corresponding targets; (**c**) shows the real-time output of the CNN+RNN+PS ML model; and (**d**) displays the FoG_condition variable.

**Table 1 bioengineering-11-01048-t001:** Distribution of datasets in PD patients.

		PD Patients				Distribution [%]		Data Characteristics	
Subject	UPDRS	Hoehn and Yahr	Gender	Medicine?	Age	FoG	Other	Num. FoG	Num. Samples
1	71	3	M	Yes	90	4.7	95.3		54.030
2	49	3	F	Yes	75	1.5	98.5	8	97.227
3	44	2	M	Yes	70	1.9	98.1	10	99.175
4	57	3	M	Yes	81	8.6	91.4	16	58.620
5	44	2	M	Yes	56	7.6	92.4	86	92.987
6	29	3	F	Yes	59	3.0	97.0	6	25.313
7	62	3	M	Yes	73	6.2	93.8	7	9.480
8	49	3	F	Yes	51	4.8	95.2	4	12.309
9	47	3	M	Yes	52	15.4	84.6	13	14.055
				Average	67	5.9	94.1	17	51.466

**Table 2 bioengineering-11-01048-t002:** Evaluation of FoG detection using a CNN+RNN+PS ML model.

Raw Dataset						CWT Dataset					
Person	Accuracy	Precision	Sensitivity	Specificity	F1	Num. PS	Accuracy	Precision	Sensitivity	Specificity	F1
1	0.941	0.010	0.006	0.951	0.016	30	0.960	0.009	0.030	0.969	0.014
2	0.986	0.008	0.005	0.994	0.007	60	0.987	0.008	0.005	0.995	0.006
3	0.959	0.024	0.018	0.982	0.020	15	0.953	0.023	0.024	0.976	0.024
4	0.913	0.050	0.067	0.934	0.057	30	0.884	0.050	0.072	0.928	0.059
5	0.947	0.024	0.031	0.971	0.026	40	0.949	0.023	0.027	0.972	0.026
6	0.958	0.023	0.022	0.980	0.021	30	0.969	0.024	0.007	0.992	0.011
Mean	0.951	0.023	0.024	0.968	0.024	34	0.950	0.022	0.027	0.972	0.023

**Table 3 bioengineering-11-01048-t003:** Sensor importance evaluation for detecting FoG.

Sensor Combination	Accuracy	Precision	Sensitivity	Specificity	F1
CF	0.910	0.051	0.057	0.943	0.053
ACC	0.911	0.050	0.062	0.937	0.056
GYRO	0.914	0.050	0.056	0.942	0.052
MA	0.912	0.051	0.057	0.943	0.054
ACC, GYRO	0.914	0.050	0.063	0.938	0.056
ACC, CF	0.915	0.050	0.059	0.940	0.054
ACC, MA	0.915	0.051	0.051	0.949	0.050
GYRO, CF	0.914	0.050	0.057	0.943	0.054
GYRO, MA	0.912	0.050	0.056	0.943	0.053
CF, MA	0.911	0.050	0.057	0.943	0.054
CF, MA, GYRO	0.913	0.050	0.063	0.937	0.056
CF, MA, ACC	0.912	0.050	0.064	0.936	0.056
MA, ACC, GYRO	0.912	0.050	0.065	0.935	0.057
CF, ACC, GYRO	0.913	0.050	0.069	0.931	0.058
CF, ACC, GYRO, MA	0.912	0.050	0.067	0.934	0.057

**Table 4 bioengineering-11-01048-t004:** Statistical analysis of the impact from vibrational actuators.

PD Patient	Vibratory Stimulation	Total Samples	FoG Samples	FoG Probability	FoG Reduction	Z-Value	*p*-Value
7	With Without	9480 16,991	586 720	0.0618 0.0424	32%	−6.47	$9.52\times 10^{-11}$
8	With Without	12,309 8412	562 217	0.0457 0.0258	44%	−7.63	$2.35\times 10^{-14}$
9	With Without	14,055 9077	2168 593	0.1542 0.0653	58%	−22.18	$2.68\times 10^{-109}$

## Data Availability

The original contributions presented in the study are included in the article/Appendix A, further inquiries can be directed to the corresponding author.

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
