# Peer review of "Wearable Online Freezing of Gait Detection and Cueing System"

_bioengineering, 2024, doi:10.3390/bioengineering11101048_

Round 1
Reviewer 1 Report
Comments and Suggestions for Authors
Thank you for submitting your paper. Lets go down the paper:
1) The abstract opens with ": This paper extends our previous work on human gait analysis ...", but most papers "stand on the shoulders of giants". Why open a paper like this? Best to just stick to what is in the current paper.
2) Correctly, the abstract gives some results, but FoG accuracy of 95.1% has been exceeded in the literature.
3) The paper presents the hardware of their system, OK, but I have two concerns then:
a) Comparing to other systems in the literature, why is this better and novel. It all looks vey standard to me. papers do not need to present what is already known or standard.
b) It seems this paper is missing may other key papers that have already presented wearable devices, and analysis of FoG in Parkinson's.
4) The CNN analysis looks good, and good to see proper ethical management of participants.
5) The paper is missing a conclusion(!)
Overall, I like the paper, but I propose to remove all the standard and known presentation (it distracts from the good novel parts), improve the literature coverage as other FoG devices are missing (I have also seen higher FoG detection results claimed), and perhaps just focus the paper around the CNN and patient participation work.
Comments on the Quality of English LanguageNot bad. No main worries.
Author Response
- The abstract opens with ": This paper extends our previous work on human gait analysis ...", but most papers "stand on the shoulders of giants". Why open a paper like this? Best to just stick to what is in the current paper.
Response: Thank you for your comment. We updated the beginning of abstract in the following way:
This paper presents a real-time wearable system designed to assist Parkinson's disease patients experiencing freezing of gait episodes. The system utilizes advanced machine learning model consisting of convolutional and recurrent neural networks, enhanced with past data sample preprocessing to achieve high accuracy, efficiency, and robustness…
- Correctly, the abstract gives some results, but FoG accuracy of 95.1% has been exceeded in the literature.
Response: You are right, that Freezing of Gait (FoG) detection accuracy rates exceeding 95.1% have been reported in the literature, and we have added new citations.
In our previous paper (https://doi.org/10.3390/s23020745), we explored nine different classifiers, and achieved FoG detection accuracy of 98,8 %. The focus of this paper is not solely on achieving the highest possible accuracy, but on developing a fully integrated system that balances detection accuracy with real-time operation, low latency, and practical deployment in real-world settings.
3.The paper presents the hardware of their system, OK, but I have two concerns then:
- a) Comparing to other systems in the literature, why is this better and novel. It all looks vey standard to me. papers do not need to present what is already known or standard.
- b) It seems this paper is missing may other key papers that have already presented wearable devices, and analysis of FoG in Parkinson's.
Response: We have expanded the introduction by including additional review articles and key papers. The primary focus of the presented paper is on a cueing (feedback) system for Parkinson's Disease (PD) patients, specifically the development of a small, efficient wearable vibrational device. This device is designed to be triggered in real-time when a Freezing of Gait (FoG) episode is detected. Additionally, we have incorporated relevant literature that addresses this specific topic..
4.The CNN analysis looks good, and good to see proper ethical management of participants.
Response: Thank you for your positive feedback regarding our CNN analysis and the ethical management of participants.
- The paper is missing a conclusion(!)
Response: Thank you, we have added chapter 5. Conclussion (page 22)
Reviewer 2 Report
Comments and Suggestions for Authors
The authors conducted a wearable online freezing of gait detection and cueing system. Some issues are needed to be fixed here:
1. Did you compare your systems with other reported literature? Please do an analysis about this.
2. As the authors described, users foot-worn sensors have some limitations. But I still suggest the authors should put your sensors under the feet since it is the most explicit way to show the episodes of FoG.
3. Basically, it is just a simple data acquisition and analysis, is there any novelty here? Have this machine learning model never been conducted in other literatures or application scenarios? Please clarify.
Author Response
- Did you compare your systems with other reported literature? Please do an analysis about this.
Response: We have included additional review references and examined other authors' research results on the accuracy of wearable systems for Freezing of Gait (FoG) detection. However, our paper focuses primarily on the vibrational cueing system designed to provide feedback to Parkinson's Disease patients. After reviewing various cueing systems, we chose vibration-based feedback due to its advantages in terms of wearability, low energy consumption, and robustness.
- As the authors described, users foot-worn sensors have some limitations. But I still suggest the authors should put your sensors under the feet since it is the most explicit way to show the episodes of FoG.
Response: In the presented paper, we added additional explanation regarding the mounting point of the IMU sensor. The choice of sensor placement is often debated, with some authors preferring foot or shoe placement, others favoring the lower leg (ankle or just below the knee), and some opting for the hip or lower back, or even a combination of multiple sensors to achieve better results. In the early phase of our research, we experimented with various sensor locations and found that placing the sensors just below the knee offered the most accurate detection of Freezing of Gait (FoG). This location not only provided the best classification accuracy in our preliminary tests but also offers practical advantages, such as easier application and greater convenience for both patients and clinicians in clinical settings. These factors led us to select this mounting point for our system.
- Basically, it is just a simple data acquisition and analysis, is there any novelty here? Have this machine learning model never been conducted in other literatures or application scenarios? Please clarify.
Response: Thank you for your valuable feedback. We appreciate the opportunity to clarify the novelty of our work and how it distinguishes itself from previous research:
- While similar machine learning models have been applied to other scenarios, the lightweight fully-personalizable model consisting of convolutional and recurrent neural networks with past data sample preprocessing has not been widely explored.
- The novelty also lies in our focus on deploying a sophisticated machine learning model in a resource-constrained environment (a microcontroller) with real-time capabilities. Many previous studies have conducted FoG detection in offline environments
- The simple past sample data preprocessing we developed allows CNN+RNN model to process broader temporal data window, resulting in good classification accuracy using just raw dataset. How much past samples to include is iteratively optimised for each patient (so another level of model personalization).
- Non-invasive muscle activity sensor is a novelty on its own.
While the architecture of our machine learning model is well-established and aligns with existing approaches, the 'Past Sample' preprocessing technique we developed is, to our knowledge, a novel contribution that has not yet been well established in the literature. This method represents a significant advancement, providing deeper insights into the data for machine learning models without introducing substantial additional processing overhead.
Round 2
Reviewer 1 Report
Comments and Suggestions for Authors
Thank you for constructively engaging in the peer review process. This resubmitted paper is much better.
I still have some comments:
1) I can clearly see the literature search has improved, I do wonder why these papers were not in your paper in the first place. However, while much improved, there is still many many papers already out there on FoG in PD, even wearable devices, that need to be considered in your. Without a full justification of the literature then novelty cannot be claimed.
2) I can see the wish to present pictures of your device, after all its a lot of work that goes into developing such devices, but my point is your Arduino device is not novel, it really is no better than others, and real healthcare device would use such technology. The paper needs to discus on what is novel. My recommendation would be to remove standard developments. However, if this was the only issue then I would propose accept. It is just that presenting such development work is counterproductive to claims of novelty.
3) I did find the paper rather difficult to read because of all the marked up comments. I cannot see what has changed. Probably best to upload a clean version, with a additional list of changes.
4) Thank you for adding the conclusion, but the conclusion opens with "This paper demonstrates the viability of a personalized, real-time wearable system for detecting and mitigating FoG episodes for Parkinson's disease patients", but such systems already exist, it has already been demonstrated.
Comments on the Quality of English LanguageNot bad.
Author Response
Thank you for constructively engaging in the peer review process. This resubmitted paper is much better.
I still have some comments:
- I can clearly see the literature search has improved, I do wonder why these papers were not in your paper in the first place. However, while much improved, there is still many many papers already out there on FoG in PD, even wearable devices, that need to be considered in your. Without a full justification of the literature then novelty cannot be claimed.
Answer: We appreciate your positive feedback on the improvement of the literature review. Regarding the omission of certain papers in the initial submission, we acknowledge that our initial review could have been more comprehensive. We have since expanded the introduction section again, offering a more detailed overview of the existing literature on FoG in PD, and have thoroughly compared wearable devices for detecting and mitigating FoG episodes. The revised version now provides a more robust justification of our work’s novelty. We also recognize that there are still key papers we might have missed. Given your expertise in the field, we would welcome any suggestions on further literature we should include to strengthen the justification and ensure we fully address the landscape.
- I can see the wish to present pictures of your device, after all its a lot of work that goes into developing such devices, but my point is your Arduino device is not novel, it really is no better than others, and real healthcare device would use such technology. The paper needs to discus on what is novel. My recommendation would be to remove standard developments. However, if this was the only issue then I would propose accept. It is just that presenting such development work is counterproductive to claims of novelty.
Answer: We appreciate your concerns regarding the use of Arduino technology (STM32 processor) and understand that these components, such as the Portenta H7, are commonly utilized in a range of advanced applications, including high-end industrial machinery, laboratory equipment, mission-critical devices, PLCs, computer vision systems, and cryptography, among others. However, the novelty of our paper does not lie in the mere use of Arduino devices, but rather in the comprehensive ecosystem we've developed. This system comprises 10 Arduino devices operating wirelessly in real time, synchronously at a speed of 40 Hz. Given the significance of this integrated approach, we believe it is essential to retain this aspect in the paper. Additionally, this work builds on our previous publication, which has been cited eight times throughout this paper.
- I did find the paper rather difficult to read because of all the marked up comments. I cannot see what has changed. Probably best to upload a clean version, with a additional list of changes.
Answer: We sincerely apologize for any inconvenience caused by the marked-up version of the manuscript. To address this, we have uploaded a clean version of the revised manuscript, which is attached. At the end of this document, all the changes made are highlighted in red. Additionally, the introduction section has been expanded to reference 52 articles.
- Thank you for adding the conclusion, but the conclusion opens with "This paper demonstrates the viability of a personalized, real-time wearable system for detecting and mitigating FoG episodes for Parkinson's disease patients", but such systems already exist, it has already been demonstrated.
Answer: We understand your point regarding the existence of systems for detecting and mitigating FoG episodes. To clarify, the novelty of our system lies in its integration of a personalized real-time approach, combining machine learning models (CNN+RNN+PS) implemented directly on a microcontroller for both detection and intervention. Additionally, our system's lightweight design and low power consumption make it suitable for long-term wearability, which sets it apart from existing devices. We revised the conclusion to better describe these aspects.

Reviewer 2 Report
Comments and Suggestions for Authors
Dear Authors,
Your manuscript could be accepted in present form.
Cheers
Author Response
Thank you for your kind feedback and for taking the time to review our manuscript again.

Round 3
Reviewer 1 Report
Comments and Suggestions for Authors
Again, than you for engaging in the review process.
1.
Looking as past point 1, yes, the literature search has improved again. I would say that it is passable at this stage. You still have many papers on FoG that you are not discussing, particularly around wavelets, and I am not going to do your literature review for you, but OK.
However, looking at your reference list - it needs a lot of work:
Sometimes paper title are capitalised, sometimes not, sometimes Parkinson is capitalised, sometimes not.
Should capitalise after ‘:’ in titles.
Why are you putting the location after some journals? A journal is a journal.
Sometimes you use et. al, and sometimes you do not and give more than the 3 authors. It is at least important to be consistent.
Ref 13 is incomplete.
Ref 35 needs to be split over two lines.
Ref 36 is missing authors
Ref 55 needs to be split over two lines.
2.
Well, it is really up to you. As I have said, discussing non state of the art tech detracts from the good parts of your paper. Being realistic, the hardware you have developed is not novel and your integrated approach is quite standard. Indeed you are incorrect in your author feedback - “advanced applications, including high-end industrial machinery, laboratory equipment, mission-critical devices, PLCs, computer vision systems, and cryptography,” - well STM32 is used in these, yes, but not Arduino.
However, it is up to you. I would not reject the paper on this basis if you are sure you want to present Arduino. Engineer to Engineer I would advise you to reflect on this for future work.
In conclusion, I recommend cleaning up the reference list throughout.
Comments on the Quality of English LanguageNot bad.
Author Response
- Looking as past point 1, yes, the literature search has improved again. I would say that it is passable at this stage. You still have many papers on FoG that you are not discussing, particularly around wavelets, and I am not going to do your literature review for you, but OK.
However, looking at your reference list - it needs a lot of work:
Sometimes paper title are capitalised, sometimes not, sometimes Parkinson is capitalised, sometimes not.
Should capitalise after ‘:’ in titles.
Why are you putting the location after some journals? A journal is a journal.
Sometimes you use et. al, and sometimes you do not and give more than the 3 authors. It is at least important to be consistent.
Ref 13 is incomplete.
Ref 35 needs to be split over two lines.
Ref 36 is missing authors
Ref 55 needs to be split over two lines.
Answer: Thank you for your feedback, we have updated the reference list as you suggested. Reference updates are highlighted below this document.
2.
Well, it is really up to you. As I have said, discussing non state of the art tech detracts from the good parts of your paper. Being realistic, the hardware you have developed is not novel and your integrated approach is quite standard. Indeed you are incorrect in your author feedback - “advanced applications, including high-end industrial machinery, laboratory equipment, mission-critical devices, PLCs, computer vision systems, and cryptography,” - well STM32 is used in these, yes, but not Arduino.
However, it is up to you. I would not reject the paper on this basis if you are sure you want to present Arduino. Engineer to Engineer I would advise you to reflect on this for future work.
Answer: I would like to clarify that the main microcontroller used in our system, the Arduino Portenta H7, features an STM32 processor and can be programmed using both the STM32 Cube IDE and Python, offering flexibility in software development. While it is true that Arduino is often associated with hobbyist projects, the Portenta H7 is designed for industrial applications, offering the same robust performance as standalone STM32 platforms.
For example, many 3D printers successfully utilize Arduino-based hardware to process millions of lines of G-code while simultaneously managing position and temperature regulation in real time. This demonstrates that, when programmed efficiently, Arduino hardware—especially higher-end boards like the Portenta H7—can deliver the necessary robustness and reliability for complex tasks, including those required in our system.
